# "Ad hoc policy decisions" in the news: Media framing analysis of a pesticide import ban in Sri Lanka

Lisa Schölin[1]*, Pramila Hashini[2], Rajaratnam Kanapathy[2], May CI van Schalkwyk[3], Michael Eddleston[1], Flemming Konradsen[4], Manjula Weerasinghe[1,2,5]

1 Centre for Pesticide Suicide Prevention, University of Edinburgh, Edinburgh, United Kingdom, 2 Faculty of Medicine, South Asian Clinical Toxicology Research Collaboration, University of Peradeniya, Peradeniya, Sri Lanka, 3 Faculty of Public Health and Policy, London School of Hygiene and Tropical Medicine, London, United Kingdom, 4 Department of Public Health, University of Copenhagen, Copenhagen, Denmark, 5 Faculty of Medicine and Allied Sciences, Department of Community Medicine, Rajarata University of Sri Lanka, Anuradhapura, Sri Lanka

* lisa.scholin@ed.ac.uk

**Data Availability Statement:** Schölin, L. (2023, August 9). Sri Lanka media analysis. Retrieved from osf.io/7kgyc.

## Abstract

Modern agriculture relies on pesticides to maximise outputs. While many highly hazardous pesticides (HHPs) are banned in places like the European Union due to concerns about harm to the environment and human health, their use continues in many low- and middle-income counties (LMICs). Pesticide suicide is a public health problem unique to countries where HHPs are used, and Sri Lanka has successfully implemented several HHP bans as part of a suicide prevention strategy. On 27 April 2021, the Sri Lankan government announced an imminent import ban of all fertilizers and pesticides which was later revoked in November 2021. The aim of this article was to explore the media framing of the import ban as it pertained to pesticides. Guided by Entman's typology of frame functions, we analysed newspaper articles from nine Sri Lankan newspapers (N = 102) between 20 April to 31 October 2021. Overall, most framings were supportive of the ban, captured by framings that articulated the ban through a health, environment, and organic farming narrative. Framings that foregrounded farmer or industry livelihoods and the pesticide market were primarily adopted to express opposition to the ban. The presence of frames opposing the ban within media articles increased over time during the study period. There was a greater proportion of opposing frames in private newspapers compared to government (78% vs 22%). Many of the articles analysed described the widespread impact of the ban, but only 11% of articles represented the voices of policy end users. This study adds understanding to the ways communication via outlets like mainstream newspapers may shape public support or opposition to pesticide bans in a LMIC. Mass media is an under-recognised factor in policy implementation and this study may inform planning to implement pesticide bans in other countries.

**Funding:** No specific funding was associated with this manuscript, but it was enabled through grants providing salary support. LS receives salary support from CPSP which is funded by a grant from Open Philanthropy, at the recommendation of GiveWell. MW, PH and KR receive salary support from a grant funded by AFSP (IIG-0-002-17). The funders had no role in study design, data collection and analysis, decision to publish, or preparation of the manuscript.

**Competing interests:** I have read the journal's policy and the authors of this manuscript have the following competing interests: LS has had research funding from the Institute for Alcohol Studies and the Carnegie Trust. MCIvS has funding through and is a co-investigator in the SPECTRUM consortium which is funded by the UK Prevention Research Partnership, a consortium of research and innovation research councils, charities, and government. ME is a WHO member of the FAO–WHO Joint Meeting on Pesticide Management and has provided technical input on WHO pesticide suicide prevention projects. ME has research funding from the UK Medical Research Council, The National Institute for Health and Care Research, and the American Foundation for Suicide Prevention (AFSP). FK also works for the Novo Nordisk Foundation, which receives funding from a range of life science companies and is a co-applicant on an AFSP grant. MW is a co-applicant on a grant funded by AFSP.

## Introduction

On 27 April 2021, the Sri Lankan government announced a ban on agrochemicals, concerning the import of all synthetic pesticides and fertilizers from 6 May 2021 (nine days later) [1]. The ban allowed continued use of all pesticides and fertilizers already in the country but prevented further import. The import ban was described as an intervention to ease pressure on foreign currency reserves and the economic impact that fertilizer subsidies had on the national financial reserves [2]. The rapidly implemented ban was expected to have significant negative consequences [2] and was subsequently revoked in November 2021.

Since the Green Revolution in the 1960s, pesticides and fertilizers have played a major role in Sri Lankan agriculture. Agriculture and farming are central to the country's economy [3] and in 2020 almost one-third of the population was employed in the agricultural sector [4]. Over time, concerns about the impact on human health and the environment from synthetic pesticides have resulted in several national bans of specific pesticide compounds. Between 1984 and 2008 Sri Lanka banned 36 highly hazardous pesticides (HHPs), including fenthion, dimethoate and paraquat due to their involvement in self-harm [5] or concerns about environmental impact [6]. Following these bans, Sri Lanka saw a dramatic reduction in the suicide rate which saved an estimated 93,000 lives between 1995 and 2015 (5). However, pesticide suicide is a global issue and an estimated one in five suicides are due to pesticide self-harm [7]. The evidence from Sri Lanka [5], and other countries indicate the success in reducing suicide rates by implementing bans [8] and are the most cost-effective intervention to reduce suicide rates [9]. It is a recommended suicide prevention strategy by the World Health Organization [10] and implementing bans does not appear to compromise agricultural yield [11–14]. However, the impact on human health has also focused on chronic diseases. Glyphosate, internationally categorised as 'probably carcinogenic to humans' by the International Agency for Research on Cancer, was banned in Sri Lanka in 2015 due to local concern that glyphosate caused chronic renal disease of unknown cause (CKDu) [15].

Mass media plays an important role in policy debates as it creates opportunities for shaping policy discourses and influencing understanding and debate among both the public and policy makers [16]. This includes defining the policy problem and the aspects of issues that are more or less important through the dissemination of particular framings [17]. For example, Katikireddi and Hilton [18] showed that both opponents and proponents of the minimum unit pricing policy on alcohol in Scotland strategically utilised mass media to influence the policy process, but compared to evidence submissions the mass media framings had more framings in opposition to the policy [18]. In Canada, an analysis of media framing of illicit tobacco trade identified that the policy responses to illicit trade were framed in ways that are more favourable to the tobacco industry [19]. In addition, the ownership of different media outlets can influence the framing of issues, which was shown in a US study where the authors concluded that "audiences of independently owned outlets are more likely to be exposed to a more robust conversation about controversial issues than their corporately owned counterparts" [20]. Public opinion can therefore be influenced by framings presented in the media about proposed policy responses to issues of importance from a public health perspective.

There is a growing body of literature exploring framings about the prevalence, causes and risk factors for chronic disease in news media, social media, marketing and entertainment [21]. Within news media, studies have identified common frames across multiple chronic conditions or risk factors; health, society, economic, practical and ideological framings [22]. There is, however, limited evidence on the framing of pesticide policy in the media. A modelling study, looking at decisions by the US Environment Protection Agency (EPA) and media

coverage, suggested that there was an effect of media coverage of pesticide risks or outcomes and the subsequent cancelling of use of pesticides on certain crops by the EPA [23].

While the media can play an important role in shaping public and policy makers' perception of during the policymaking process, the current study aimed to explore the media framings of the 2021 import ban in Sri Lanka, focusing on pesticides, in response to its abrupt announcement and during its rapid implementation. Our objectives were to: i) identify and describe the framings used in Sri Lankan newspaper articles relating to the import ban, and ii) explore whether there were any differences in frames used in articles published in newspapers with different ownership (government or private).

## Materials and methods

### Study design

In this study we aimed to explore how the import ban was framed in Sri Lankan newspaper articles between 20 April and 31 October 2021. We focus solely on framings about the banning of pesticides due to its relevance at a global level as a suicide prevention strategy and our interest in interventions to reduce harm from pesticide poisoning [24]. The study period covered the announcement of the ban, the promulgation of the law, and the two main agricultural seasons (Fig 1). It did not, however, include the revocation of the ban as this fell outside of the period of data collection.

### Search strategy

We searched nine national online newspapers in Sinhala (Lankadeepa, Dinamina, Diwaina), Tamil (Veerakesari, Thinakaran, Thinakural) and English (Daily Mirror, Daily News, Sunday Times) (Table 1) [25]. All newspapers were searched with the key terms 'national pesticide ban', 'pesticides', 'fertilizer', 'agrochemicals', 'agrochemical ban' or 'organic farming' in Sinhala, Tamil, and English. Keywords were informed by discussions with local researchers and, notably, we used the keyword 'fertilizer' to identify all potentially relevant articles that discussed the (combined) import ban but screened out articles that focused exclusively on the banning of fertilizers. The article did not need to focus exclusively on the ban and could mention other related issues or topics. For example, articles on a relevant topic such as pesticide shortages or difficulties with crops could contain information about the ban and these were included.

### Data analysis

Our analytical approach was guided by the concept of framing, which focuses on examining how an issue is constructed to create a particular problem definition, and to persuade target audiences of its significance and what are to be understood as legitimate solutions [17]. This includes both prominence of messages within a text and the omission of certain information, or other perspectives and interpretations, for an issue to be given more (or less) salience [17]. Entman [17] proposed a typology of frame functions that captures how frames can be employed to i) describe problems, ii) diagnose causes, iii) make moral judgements, and iv) offer possible solutions.

Our analysis adapted the approach used by Atanasova and Koteyko [26], in which we focused on identifying frames and their articulation of problems, their causes, consequences, and proposed solutions. These sub-elements are core to frame arguments and use of this approach to framing analysis enabled a structured approach that could be used consistently across multiple coders. The coding framework was developed by LS and MW through iterative

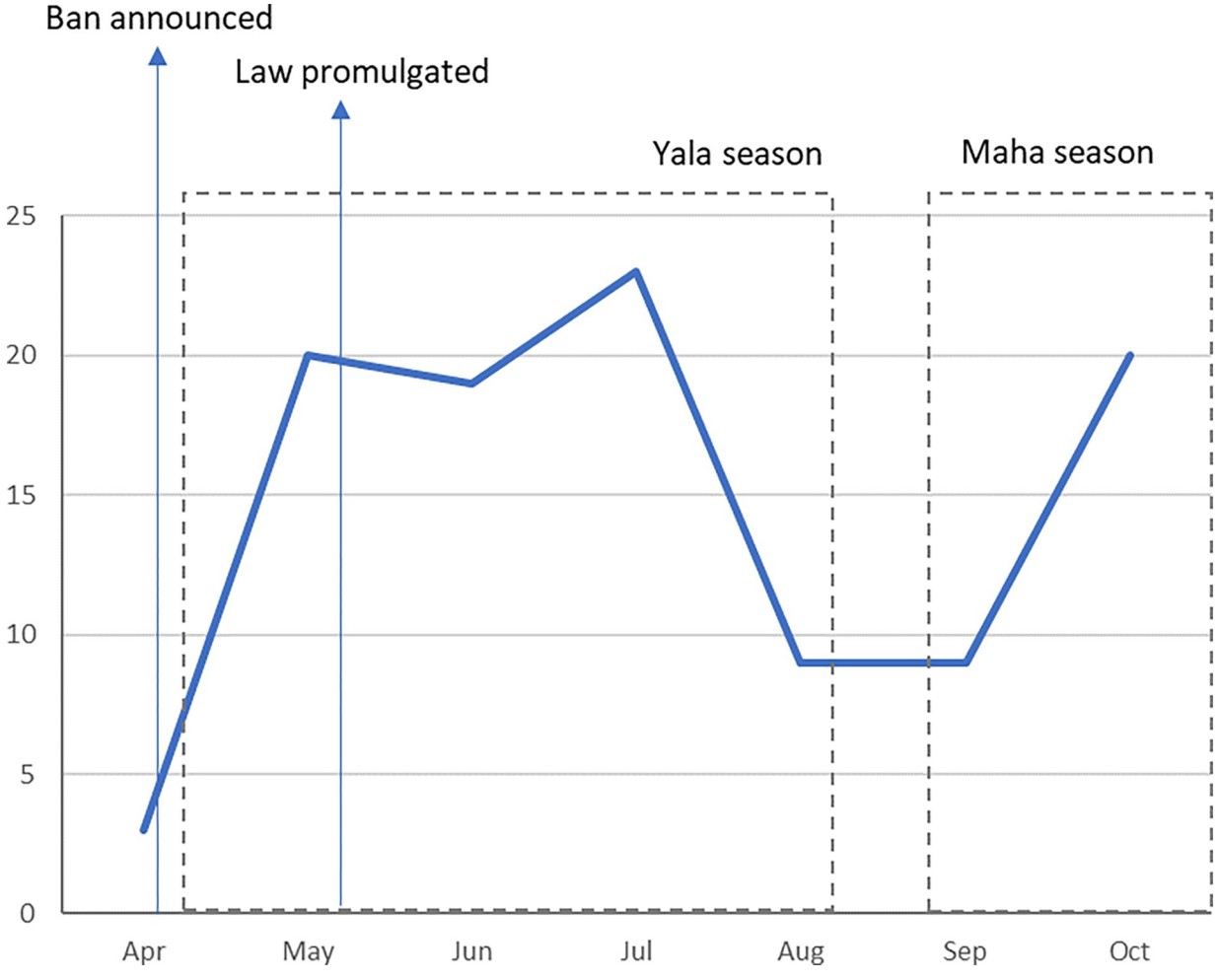

**Fig 1. Number of articles published each month of the data collection period.**

**Table 1. Characteristics and numbers of included newspaper articles by newspaper.**

| Newspaper | Frequency | Circulation[a] | Language | Ownership |
|---|---|---|---|---|
| Dinamina | Daily | 75,000 | Sinhala | Government |
| Lankadeepa | Daily<br>Sunday | 250,000<br>560,000 | Sinhala | Private |
| Diwaina | Daily<br>Sunday | 156,000<br>340,000 | Sinhala | Private |
| Veerakesari | Daily | 140,000 | Tamil | Private |
| Thinakaran | Daily | 120,000 | Tamil | Government |
| Thinakural [b] | Daily | | Tamil | Private |
| Daily Mirror | Daily | 76,000 | English | Private |
| Daily News | Daily | 88,000 | English | Government |
| Sunday Times | Weekly | 330,000 | English | Private |

[a]Source: Wikipedia [25]

[b]No information about circulation available.

coding, using a subsample of English newspaper articles which were double coded in stages and discussed to develop an initial coding framework. In total, 69% of English articles were double coded by LS and MW, and LS then coded the remaining articles independently. The initial coding framework was discussed between LS, MW, PH and RK. MW and PH coded 10 (16%) of Sinhala articles which were also translated into English by an official translator and triple coded by LS. Final revisions were then made to the coding framework and applied to all remaining articles. All Sinhala articles were double coded by MW and PH. RK coded all Tamil newspaper articles and a subsample of three articles (21%) were translated to Sinhala and double coded by MW. Further translation and double coding were not feasible within the project. No evaluation of coding agreement was undertaken. The identified frames and frame elements which formed the coding framework are presented in S1 Table.

We recorded basic characteristics of each article including date, language, type of article, source, and voices (i.e. individuals quoted within the article) represented in an Excel spreadsheet and performed descriptive statistics. We also calculated the frequency of each frame and its position (supportive, in opposition, or neutral i.e. neither critical of the policy decision nor supportive), and the distribution of particular framings by newspaper ownership.

## Results

### Characteristics of newspaper articles

Our initial search resulted in 699 newspaper articles which after screening left 102 relevant articles. Table 2 describes the characteristics of included articles. Most articles were in Sinhala, about two thirds were from privately owned newspapers, and the newspaper with the greatest number of articles was Diwaina. In around half of all articles, opinions of different stakeholders were presented and in the 61 articles where stakeholders were represented through direct quotes, just over half featured policy makers while only eleven articles (11%) included policy end users.

We explored the number of articles published each month over the six-month period (Fig 1). The number of articles increased after the ban was announced in April, with the highest number in July, as the Yala (May to August) harvest season was coming to an end and another increase in October after the Maha (September to March) harvest season had started.

### Framing of the pesticide ban

We identified eight different framings of the import ban in the 102 included articles analysed for the study. These framings were used a total of 139 times, with each of the framings differing in the number of times they were identified: human health (22%), organic farming (21%), farmer livelihoods (17%), environmental impact (14%), industry livelihoods (10%), pesticide market (7%), food security (5%), and political decisions (3%) (Fig 2). Of these, 55% (n = 76) were supportive of the ban, 40% (n = 55) were opposed to the ban, and 6% (n = 8) were neutral. We identified that the same frame theme was employed in both support of and against the ban. However, most of the health, environment, and organic farming frames were supportive of the ban while opposition was dominant among farmers or industry livelihoods, and pesticide markets. There was an increase in frames in opposition over the study period (Fig 3) and supportive framings declined after July (during the Yala harvest season).

In the 30 articles that were identified as presenting multiple framings, the most common combination of framings was environmental impact and human health, followed by organic farming and human health, and organic farming and environmental impact (Fig 4). These combinations of framings were mainly the co-presentation of two supportive framings (50%) or two opposition framings (27%) though we also saw supportive and opposed framings used

**Table 2. Characteristics of included newspaper articles.**

| Characteristic | Grouping | n (%) |
|---|---|---|
| Language | Sinhala | 61 (60) |
| | English | 28 (27) |
| | Tamil | 13 (13) |
| Ownership | Government | 38 (37) |
| | Private | 64 (63) |
| Newspaper | Diwaina | 29 (28) |
| | Dinamina | 17 (17) |
| | Lankadeepa | 15 (15) |
| | Daily News | 12 (12) |
| | Sunday Times | 10 (10) |
| | Thinakaran | 9 (9) |
| | Daily Mirror | 6 (6) |
| | Veerakesari | 3 (3) |
| | Thinakural | 1 (1) |
| Type of article | News | 68 (67) |
| | Feature | 17 (17) |
| | Editorial | 10 (10) |
| | Column | 4 (4) |
| | Letter to editor | 2 (2) |
| | Other | 1 (1) |
| Stakeholders represented[a] | None | 49 (48) |
| | Policy makers[b] | 34 (33) |
| | Other[c] | 12 (12) |
| | Policy end users[d] | 11 (11) |
| | Academic | 5 (5) |

[a]Not mutually exclusive

[b]Government and parliament representatives, president

[c]e.g. environmentalists, agriculture expert, import expert

[d]Farmers, plantation owners, farmer federation and industry representatives.

together within the same article (17%) and one article presented a supportive frame with a natural frame.

## Impact on health

The most common frame identified was human health, which focused on narratives of how synthetic agrochemicals negatively influence the health of the Sri Lankan population by, for example, contributing to development of noncommunicable diseases (NCDs) such as CKDu. Supportive frames drew a link between pesticides and NCDs: "people who were in debt to take pesticides for paddy field went to take medicine for kidney disease and became more in debt. The number of people dying from non-communicable diseases has increased, not decreased" [27]. Many articles were less specific about the impact on health; one article interviewed an agricultural instructor who stated that "the use of agrochemicals is responsible for most of the diseases in our society today" [28]. The use, or overuse, of synthetic pesticides was described as an underlying cause that necessitated bans in all articles using the health frame in support of the ban. But use of pesticides was also used in opposition to the ban; one article claimed that "most pesticides used in organic farming are as poisonous as chemical pesticides" [29]. This

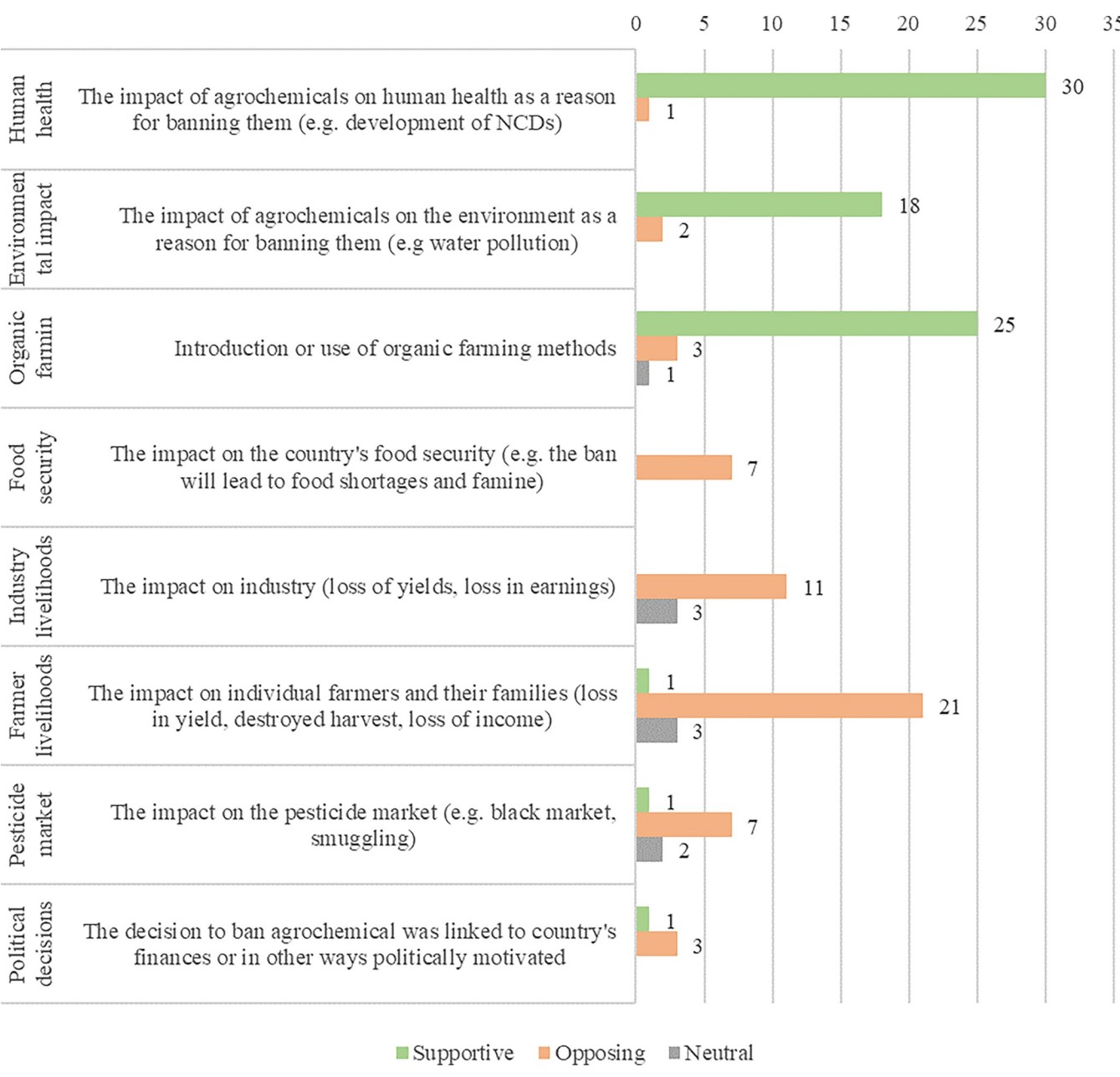

**Fig 2. Position of identified framings.**

article positioned the issue as improper use of synthetic agrochemicals rather than their toxicity and impact on human health. Organic farming was seen as the solution in most human health frames, alongside measures to reduce availability to synthetic pesticides, and argued that synthetic pesticide use "can gradually be overcome through the application of [. . .] alternative pest control strategies" [30]. Another article cited the then President, who stated that "my government's decision to ban the importation of synthetic agrochemicals will help meet the long-term need to migrate to healthier and more environmentally sustainable organic farming systems" [31].

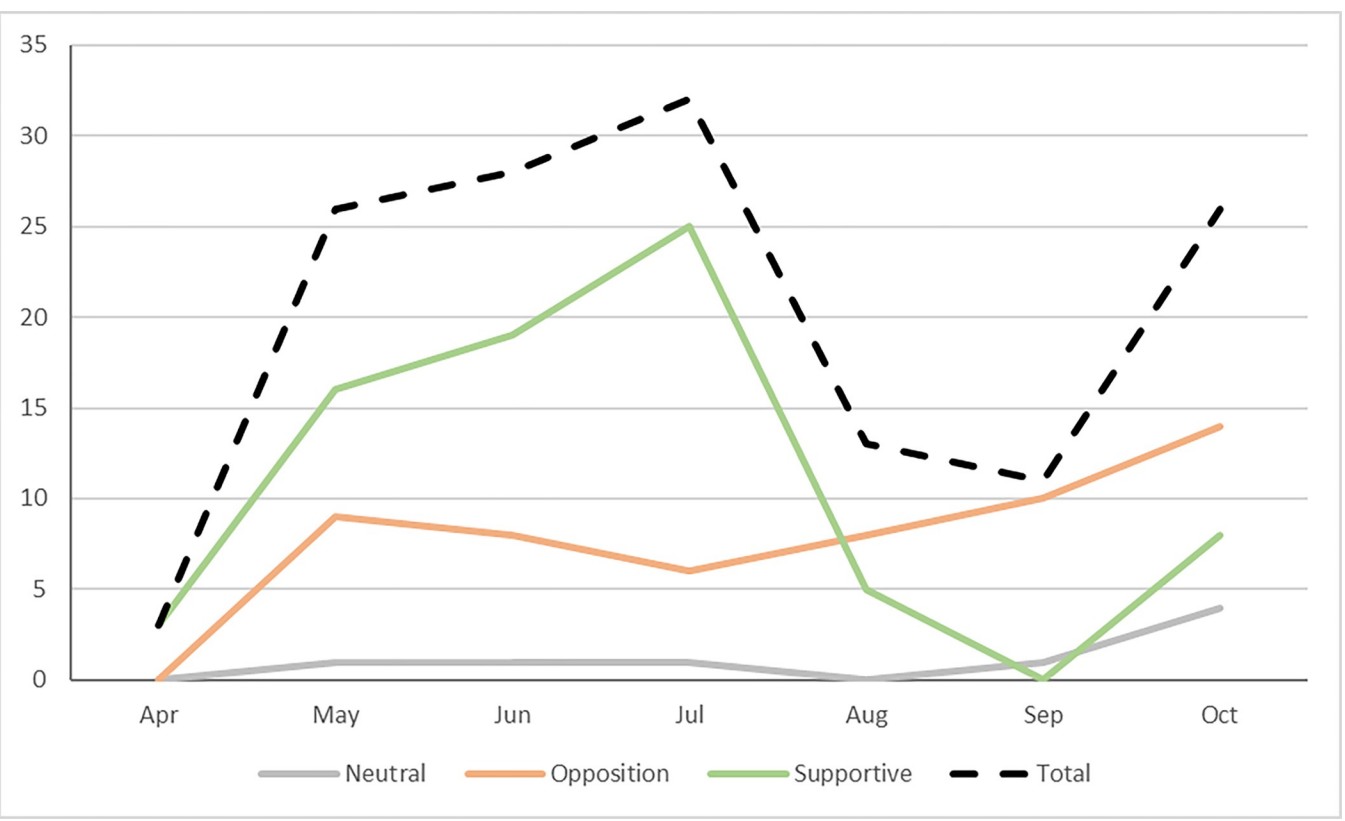

**Fig 3. Frame position by month.**

## Organic farming

The other frequently adopted framing focused on banning pesticides as part of the greater vision of transforming Sri Lanka's agriculture to being exclusively organic [32]. Support for the ban focused on use or overuse of pesticides as a key cause of environmental and human health issues. In the small number of articles in opposition of the ban, this frame was employed to highlighted challenges to implementing organic farming, which has "devastating repercussions and inevitable food security issues [...] perhaps leading to a famine in the country" [33]. Moving towards organic farming was one of the solutions presented within this frame, along with education of farmers and modern technology. Importantly, this frame fit within a wider agenda in Sri Lanka of transitioning towards organic agriculture which predominantly featured in the news debate of the import ban.

## Farmer livelihoods

The farmer livelihoods frame focused on how the ban impacted farmers, primarily framed in opposition of the ban with problems stemming from shortages of pesticides: "thousands of farmers across the country are struggling without adequate chemical fertilizers or from a shortage of agro-pesticides" [34]. One article reported on farmer protests and described how farmers "left the vegetable cultivation in the Maha season and their family members are in dire straits due to the loss of their income" [35]. Shortage of pesticides was also described as a cause of the problem in neutral frames, but these were short articles that simply described events without any emphasis that indicated a position towards the ban. The consequences of banning pesticides were described as having significant impact on farmers, leading to disappointment,

**Fig 4. Combinations of framings in articles with more than one frame.**

destroyed crops, and financial losses. Public expressions of these frustrations were described, such as a "Dambulla farmer whose brinjal harvest was destroyed due to lack of agrochemicals set fire to the harvest publicly" [36]. The solution to the problems resulting from the ban was for the government to make pesticides available. The Opposition Leader requested that the President "take immediate steps to provide the farmer communities with adequate fertilizer, weedicide and pesticide without further destroying the country's agricultural sector" [37]. Another article described how farmers "requested the government to provide chemical fertilizers and agro-chemicals" for the Maha season [35]. A few articles suggested other solutions including private-public partnerships, focusing on developing new technologies and monitoring, and reverting the ban all together. It was noted in one article that successful implementation of the ban should have included consultation with farmers, which is why reverting the ban was described as a necessary action given the proven impact the ban had [38].

## Environmental impact

Like the human health frame, within the environmental impact frame, the problem was presented as stemming from the availability, use and overuse of synthetic pesticides. The impact was often described in general terms: "[. . .] the excessive use of chemical fertilizer together with weedicides and pesticides has led to a series of environmental and health hazards" [37]. While almost all frames were supportive of the ban, one article framed environmental impact within opposition to the ban. It described that "the environmental pollution due to excessive usage takes place due to lack of knowledge due to poor agricultural extension services in the country" [33]. Overall, the excessive use of pesticides and other agrochemicals was described as leading to soil infertility, pollution of waterways, and emission of greenhouse gases. The solution was presented as moving to organic agriculture, with the ban described as "essential for long-term eco-friendly organic agriculture" [39]. More specifically, one article described the ban as a "powerful and complicated decision" by the President to implement the ban as a "first step to limiting the use of chemicals in agriculture" [40].

## Industry livelihoods

The industry livelihood, similarly to the farmer livelihood frame, focused on the impact on industry profit and sustainability of production without agrochemicals. Specific industries growing crops important to Sri Lanka's economy such as tea, coconut, rubber, and ornamental flowers, were featured. The problem resulting from the ban was, again, described as resulting in shortages of pesticides and the "ad hoc decisions" the government had taken "without seeking the views of plantation stakeholders" would have "major negative impact for the industry" [41]. The ban and subsequent shortages in agrochemicals were described as impacting on yields and quality of crops. In two articles, this frame argued that "if the production of paddy decreases to 40%, if it decreases to 60% from vegetables, a famine will arise in the country by next year" and "there is a food shortage in the country due to the agrochemical ban which has no scientific basis" [42]. Articles that described solutions focused on making pesticides available, though one article suggested the solution was to allow import of agrochemicals for particular crops, such as tea [43], and one described that phasing out products "should be done systematically" rather than a complete ban [42]. However, in half of all instances where this frame was used no solutions were presented.

## Pesticide market

The pesticide market frame, which featured infrequently, focused on how the ban had unintended consequences that the shortage of pesticides was creating. Articles described that an

"insecticide packet which was 700 rupees is sold for 1700 rupees" [44] and that "with the government ban on import of chemical inputs, some traders are taking advantage of this opportunity to hoard fertilizer and pesticides in their shops for profit" [38]. The shortage of pesticides and herbicides was also described as leading to "smuggling of substandard agrochemicals brought by sea route into the country" [45] and "creating a black market for such agrochemicals and smuggling of these banned items" [46]. The latter article further described the situation that had been created as a representative of a farmer federation described how "farmers now depend on the black market to buy agrichemicals paying Rs. 12,000 for a packet or bottle of pesticides priced Rs, 6,000 previously" [46]. Only three instances of this frame presented solutions to the problems the ban had created, with two proposing that pesticides need to be made available. One article described raids in market to address smuggled products, yet this article did not present a broader long-term solution for the problems created by the imposed import ban.

## Food security

In the food security frame, also infrequently featured, the ban was seen within a wider issue of food security for the population of Sri Lanka and significant consequences the lack of agrochemicals had, or could have, on the ability of feeding the nation. One article, which positioned the ban within the wider ambition of Sri Lankan agriculture to become fully organic, stated that the "stampede to return to agriculture's roots of using nothing more than nature's compost has remained reined but the stark prospect of a greater evil ensuing in its wake: the dismal spectre of worldwide famine due to worldwide food shortages" [36]. Another article expressed that "without agrochemicals the paddy fields have already started to decay" and that the government was "staging the mega-drama 'vision of famine' in the maha season" [47]. Many of the articles where this frame was used discussed how the speed with which the ban was implemented, even if the article in some way supported the idea of agriculture free from synthetic pesticides, was a factor that would lead to unintended consequences.

## Political decisions

Perhaps surprisingly, only four instances were coded where the ban was framed as a political decision to deal with the country's emerging financial crisis. Foreign currency reserves were seen as the cause of the problem in the three frames in opposition to the ban and the one in support of the ban. One article stipulated, in contrast to the many instances of the organic farming frame that was coded across multiple articles, that "imports require the government to spend more of the nation's now dwindling foreign reserves" [48] and another article simply stated that the ban was a "demonstration of the government's incompetence" [49]. These decisions, albeit to save money, were described as leading to food shortages and crops being destroyed since no agrochemicals were available to support the harvest season. Only one of the articles where the political decision frame was used proposed a solution. The author of that article, who was a former minister for agriculture, stated that: "In order to save lives, and prevent the decimation of livelihoods, we call upon the government to reverse their decision immediately. Do the sensible thing: lift the ban on imports of agrochemicals" [48].

## Framing by newspaper type

We considered whether the framings and support of the ban differed by ownership of the newspaper. There was a total of 55 instances of distinct framings within the 38 government-associated newspaper articles and 84 within the 64 articles from private newspapers. There was a greater proportion of opposing frames in private newspapers compared to government (78%

vs 22%), while a smaller proportion of supporting frames were from private newspapers (45% vs 55%). The neutral frames were predominantly from private newspapers; however, it is worth noting that only nine frames were neutral. The position of each frame within government and private newspapers was relatively similar, with a few exceptions. All political decisions and industry livelihood frames in government newspapers were in opposition of the ban while in private newspapers there were also some supportive and neutral frames. While almost all farmer livelihood frames in private newspapers were in opposition, government articles also had neutral and supportive frames.

## Discussion

In this study we analysed newspaper coverage of the rapidly implemented import ban on agrochemicals in Sri Lanka in 2021, to identify and describe the framings of the ban as it pertained to pesticides. Our paper thus provides additional insights into media framings by focusing on policy developments surrounding an abrupt policy announcement and rapid implementation, which differs from previous analyses that focus on more protracted policy debates and policy-making processes that culminate (sometimes after years of deliberation) in final policy decisions [50]. Following the announcement of the ban, newspaper coverage was to a greater extent supportive of the ban, yet this study cannot draw any conclusions on how these framings may have affected public opinion. The ban was mainly described in supportive terms, framed as benefiting human health while fitting into the aspiration for transitioning towards organic farming methods in the country. On the other hand, many articles included framings opposing the ban and describing the negative impact on livelihoods of farmers and industry and unintended consequences to the pesticide market. The lack of consensus on the reasons behind the ban may illustrate the challenges that swift implementation of such bans may have and could serve as an important example for other countries or regions where policy makers, advocates and researchers seek to garner public support for such bans to protect human health and the environment. While government-owned newspapers mostly lacked opposing frames, supportive frames had an almost even split between private and government-owned newspapers. This differs to other research, such as reporting of the Colombo City Port which is developed by a Chinese company, in which TV news reporting by government-owned channels were supportive of the initiative [51]. Discourse can however vary across media spaces, beyond the binary division of private and government-owned newspapers. Kaiser [52] explored pesticide legitimacy in Switzerland through different media sources (mainstream media and farming press) and found that mainstream media focused on harmful consequences of pesticides while the farming press focused on solutions that legitimised pesticide use, such as the use of technology to reduce excessive pesticide use. Overall, however, the author highlighted that the debates over an eleven-year period worked to stabilise discourses around the legitimacy of pesticide use and advocating away from regulatory approaches.

The overall supportive sentiment towards the ban resonates with a review by Rowbotham et al. [22], which found that media coverage across multitude policy interventions (alcohol, tobacco, diet and nutrition), in almost exclusively high-income countries, was supportive [22]. This might reflect the history of pesticide bans on health and environmental grounds [5,6] and a wider aspiration and subsequent discourse of organic farming as the future of Sri Lankan agriculture. Organic farming is part of Sri Lanka's agricultural policy and in 2019, 2.5% of Sri Lanka's agricultural land was cultivated organically, compared to the global average of 1.5% [53]. The supportive framings within health, environment and organic farming may therefore signal that this is a wider discourse in the country, while the opposing frames indicating the realities of a swiftly implemented ban among policy users, whose voices were largely absent in

newspaper coverage. Further research using different methods is needed to understand how different stakeholders, such as industry, advocates, and farmers, seek to influence pesticide policy in Sri Lanka and with what impacts.

Within the articles we analysed, the reason for the ban was not clearly stated and only a small number of articles included framings relating to political decisions, specifically the financial expenditure of agrochemicals [2]. Sri Lanka has a long history of subsidizing fertilizers for farmers, which had benefits to food production and export but simultaneously putting pressures on the country's expenditure on agriculture [2]. Our focus on pesticides might reflect the lack of political decision frames, as the subsidies provided for fertilizers might mean that this framing would be more common in articles that specifically focused on the import ban of fertilizer. Another motive was the impact from overuse of chemical fertilizers [2], which was also reflected in our sample of articles focusing on pesticides. Use or overuse was the most common cause that necessitated the ban (48% of all causes coded), but there was significant overlap with discussions also including fertilizers and it was not always clear whether the article was talking about agrochemicals in general, or fertilizers or pesticides specifically. An analysis of articles focusing on fertilizers will therefore be informative to explore if framings for different types of agrochemicals differ.

Framings function to shape how an issue is given more or less salience by foregrounding particular aspects, narratives and understandings, while concealing others [17]. In our study, the absence of voices of policy end users (farmers and pesticide vendors) will have influenced the framings presented in the articles included in the analysis and may have contributed to establishing an overly positive view of the pesticide ban. Notably, of those framings presenting by end users, two fifths were identified as being in opposition to the ban. Frame omission is an important part of framing an issue [54] and within media framing, this is illustrated in, for example, news reports about illicit drug use where public health, harm reduction and decriminalization frames are absent which may result in more stigmatizing perceptions of people who use drugs [55]. Similarly, a study on corporate social responsibility activities of tobacco companies indicated an absence of stakeholders speaking on tobacco control, which was noted as a "missed opportunity to influence public and policy maker opinion" (p. 7) [56]. The omissions of certain stakeholder voices, and certain framings, within newspaper articles may therefore have had an impact on public opinion about the ban. The current study cannot, however, draw any conclusions about what that impact might have been, but it is notable that supportive frames decreased over time and opposing frames increased leading up to the revocation of the ban.

## Strengths and limitations

This study included articles from nine of the most widely read newspapers in Sri Lanka and covered a six-month period including the announcement and implementation of the ban. The coding process was highly collaborative and ongoing discussions ensured that the coding framework was applied consistently across articles in all three languages. However, this study has several limitations that should be acknowledged. Not all data were double coded, and no inter-rater reliability was recorded; however, most articles were coded by two researchers and the development of the coding framework was through ongoing discussions within the research team. The study only included online versions of newspapers, however as the content in online and printed newspapers is the same, we do not believe anything will have been missed by using online versions. Newspapers were selected based on their circulation and differences between government and private newspapers are limited due to the small number of government newspapers in Sri Lanka. The ban was reversed after the end of our data collection

period and the framing after the reversal might have changed but it was out of scope for our study to explore this. Nevertheless, this study adds understanding to the ways communication via outlets like mainstream newspapers may shape public support or opposition to pesticide bans in a LMIC. Mass media is an under-recognised factor in policy implementation and this study may inform planning to implement pesticide bans in other countries. Further research is needed to explore how the rapid implementation and subsequent reversal of this ban was experienced by key stakeholders and what impact different framings within newspaper coverage might have had on different target groups (the public, policy makers, farmers, and pesticide vendors).

## Supporting information

**S1 Table. Frame sub-elements.**
(DOCX)

## Acknowledgments

The authors would like to thank Sanduni Peiris and Chathuranga Piyasena for their assistance with double checking articles during the screening process.

## Author Contributions

**Conceptualization:** Lisa Schölin, Manjula Weerasinghe.

**Data curation:** Lisa Schölin, Pramila Hashini, Rajaratnam Kanapathy, Manjula Weerasinghe.

**Formal analysis:** Lisa Schölin, Pramila Hashini, Rajaratnam Kanapathy, Manjula Weerasinghe.

**Methodology:** Lisa Schölin, May CI van Schalkwyk, Manjula Weerasinghe.

**Supervision:** May CI van Schalkwyk, Michael Eddleston, Flemming Konradsen, Manjula Weerasinghe.

**Writing – original draft:** Lisa Schölin, Manjula Weerasinghe.

**Writing – review & editing:** Lisa Schölin, May CI van Schalkwyk, Michael Eddleston, Flemming Konradsen, Manjula Weerasinghe.

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
