## [Decision Letter · Decision Letter 0]

8 Mar 2024

PGPH-D-23-01738

“Ad hoc policy decisions” in the news: media framing analysis of a pesticide import ban in Sri Lanka

Dear Dr. Scholin,

Thank you for submitting your manuscript to PLOS Global Public Health. After careful consideration, we feel that it has merit but does not fully meet PLOS Global Public Health’s publication criteria as it currently stands. Therefore, we invite you to submit a revised version of the manuscript that addresses the points raised during the review process.

We look forward to receiving your revised manuscript.

Kind regards,

Prashanth Nuggehalli Srinivas, MBBS, MPH, PhD

Academic Editor

Journal Requirements:

Additional Editor Comments (if provided):

Reviewers' comments:

Reviewer's Responses to Questions

**Comments to the Author**

1. Does this manuscript meet PLOS Global Public Health’s publication criteria? Is the manuscript technically sound, and do the data support the conclusions? The manuscript must describe methodologically and ethically rigorous research with conclusions that are appropriately drawn based on the data presented.

Reviewer #1: Yes

Reviewer #2: Yes

2. Has the statistical analysis been performed appropriately and rigorously?

Reviewer #1: N/A

Reviewer #2: Yes

3. Have the authors made all data underlying the findings in their manuscript fully available (please refer to the Data Availability Statement at the start of the manuscript PDF file)?

Reviewer #1: Yes

Reviewer #2: Yes

4. Is the manuscript presented in an intelligible fashion and written in standard English?

Reviewer #1: Yes

Reviewer #2: Yes

5. Review Comments to the Author

Reviewer #1: I find the article informative and generally well executed and well-written.

This is a good example that value can be extracted in studies without having to resort to complex statistical analysis, provided researchers are still willing to do the necessary work.

The approach taken, and the development of the coding framework, all seem well executed, and the limitation of not assessing the inter-rater reliability of the coding (primarily due to not all the coding being done in duplicate) is already acknowledged, but while this is a weakness, it does not render the results implausible.

I can make some small recommendations:

1. Results (first paragraph). The table indicates 11 articles for policy end users, and not ten.

2. PLOS uses ICMJE referencing. According to ICMJE guidelines, foreign language sources should be referenced in the original language, with an (optional, but I think recommended) translation in brackets. See https://www.nlm.nih.gov/bsd/uniform_requirements.html

This also might clarify the odd title to reference 35, which reads "...there will be famine and famine in the country..." which I assume should be "food shortage and famine".

There are also typos in references 2, 32, 37.

3. There is a small language error in Figure 2, where I suspect it should read "the impact of the ban on the pesticide market..." instead of "the impact of the ban has on the pesticide market"

4. Environmental impact section: the last sentence needs a citation.

5. Pesticide market section: Delete the word 'how' from "One article described how raids in market to address smuggled products, yet this article did not present a broader long-term solution for the problems created by the imposed import ban."

6. Political decisions section: Recommend changing 'one' to 'that' in the sentence "The author of one article, who was a former minister for agriculture, "

The only larger recommendation I could make is that I think the authors missed a key point regarding the fact that many articles appear to have included multiple framings (They note: "We identified eight different framings of the import ban in the 102 included articles analysed for the study. These framings were used a total of 139 times, with each of the framings differing in the number of times they were identified."). I suspect the combination of framings itself would also have proved insightful, especially in light of the fact that some framings were used both in support of, and in opposition to, the ban. Perhaps, the combinations of framings in articles supporting or opposing the ban could have revealed a pattern? Also, I think a heat map showing which framings were used in combination would also be insightful, and may add further depth to the analysis the authors have undertaken.

Reviewer #2: This manuscript explores a relevant issue and I read it with great interest. The authors empirically analyse the use of frames concerning a ban of pesticide imports in Sri Lanka in 2021. They use a straightforward methodological approach to analyse a total of 102 media articles. The findings show a shift from the use of frames predominantly supporting the ban to more frames in opposition to the ban.

I have very little critique and would be happy to see this published as is. My only disappointment was that the comparison of frames used by governmental vs. private newspapers appears quite prominently in the second part of the research question and thus in the results. However, the argument for drawing this comparison is stated rather implicitly in the introduction. I would suggest to make this a bit more explicit and add a small paragraph about (specifically these) different newspaper types.

Similarly, the issue of pesticide suicide – which seems to be a primary motivation for conducting this research – is merely introduced in passing (second paragraph in the introduction). A few more words on this would help a reader not familiar with the topic.

Some very minor points.

First, in the introduction, the statement “[…] implementing bans does not appear to compromise agricultural yield (9)” could maybe be supported by some additional references.

Second, in the discussion, the authors state that “[…] it is notable that supportive frames decreased over time and opposing frames increased leading up to the revocation of the ban” (p. 15). If possible, I would be interested to read whether it can be assumed that these voices created pressure that led to the government’s decision to revoke the ban, or whether this was rather associated with an indecisiveness/ other political considerations on the part of the government. (In a very different context, this paper https://doi.org/10.1016/j.eist.2023.100777 offers an analysis of discursive struggles in the media and the (re)creation of pesticide legitimacy.)

Third, the section “Discussion” may also be entitled “Discussion and conclusion”, in my opinion.

Last, the manuscript entails a few typos (e.g., in this sentence on p. 4: “There is a growing body of literature exploring framings about the of prevalence…”) which the authors may wish to correct in a final round of proofreading.

6. PLOS authors have the option to publish the peer review history of their article (what does this mean?). If published, this will include your full peer review and any attached files.

**Do you want your identity to be public for this peer review?** For information about this choice, including consent withdrawal, please see our Privacy Policy.

Reviewer #1: No

Reviewer #2: No

---

## [Decision Letter · Decision Letter 1]

28 May 2024

PGPH-D-23-01738R1

“Ad hoc policy decisions” in the news: media framing analysis of a pesticide import ban in Sri Lanka

Dear Dr. Scholin,

Thank you for submitting your manuscript to PLOS Global Public Health. After careful consideration, we feel that it has merit but does not fully meet PLOS Global Public Health’s publication criteria as it currently stands. Therefore, we invite you to submit a revised version of the manuscript that addresses the points raised during the review process.

Please carefully copy-edit for minor grammatical errors including the ones pointed out below by one of the reviewers. The paper appears to be otherwise ready to move to accept, save for minor copy-edits. 

We look forward to receiving your revised manuscript.

Kind regards,

Prashanth Nuggehalli Srinivas, MBBS, MPH, PhD

Academic Editor

Journal Requirements:

Additional Editor Comments (if provided):

First, it seems as if the typo in Figure 2 still remains ("the impact of the ban has on the pesticide market" is not grammatically correct, and I suggested "the impact of the ban on the pesticide market". The authors noted in their responses that they had corrected this, so perhaps they uploaded the old version when doing their resubmission.

Second, on p16, "will have influence the framings" should be "will have influenced the framings"

Reviewers' comments:

Reviewer's Responses to Questions

**Comments to the Author**

1. If the authors have adequately addressed your comments raised in a previous round of review and you feel that this manuscript is now acceptable for publication, you may indicate that here to bypass the “Comments to the Author” section, enter your conflict of interest statement in the “Confidential to Editor” section, and submit your "Accept" recommendation.

Reviewer #1: All comments have been addressed

Reviewer #2: All comments have been addressed

2. Does this manuscript meet PLOS Global Public Health’s publication criteria? Is the manuscript technically sound, and do the data support the conclusions? The manuscript must describe methodologically and ethically rigorous research with conclusions that are appropriately drawn based on the data presented.

Reviewer #1: Yes

Reviewer #2: Yes

3. Has the statistical analysis been performed appropriately and rigorously?

Reviewer #1: N/A

Reviewer #2: Yes

4. Have the authors made all data underlying the findings in their manuscript fully available (please refer to the Data Availability Statement at the start of the manuscript PDF file)?

Reviewer #1: No

Reviewer #2: Yes

5. Is the manuscript presented in an intelligible fashion and written in standard English?

Reviewer #1: Yes

Reviewer #2: Yes

6. Review Comments to the Author

Reviewer #1: I thank the authors for taking our feedback on board, and commend them for a well written and informative paper.

Two minor points do need attention:

First, it seems as if the typo in Figure 2 still remains ("the impact of the ban has on the pesticide market" is not grammatically correct, and I suggested "the impact of the ban on the pesticide market". The authors noted in their responses that they had corrected this, so perhaps they uploaded the old version when doing their resubmission.

Second, on p16, "will have influence the framings" should be "will have influenced the framings"

Reviewer #2: (No Response)

7. PLOS authors have the option to publish the peer review history of their article (what does this mean?). If published, this will include your full peer review and any attached files.

**Do you want your identity to be public for this peer review?** For information about this choice, including consent withdrawal, please see our Privacy Policy.

Reviewer #1: **Yes: **Jacques Eugene Raubenheimer

Reviewer #2: No

---

## [Editor Report · Decision Letter 2]

26 Jun 2024

“Ad hoc policy decisions” in the news: media framing analysis of a pesticide import ban in Sri Lanka

PGPH-D-23-01738R2

Dear Dr Scholin,

We are pleased to inform you that your manuscript '“Ad hoc policy decisions” in the news: media framing analysis of a pesticide import ban in Sri Lanka' has been provisionally accepted for publication in PLOS Global Public Health.

Best regards,

Prashanth Nuggehalli Srinivas, MBBS, MPH, PhD

Academic Editor